



# Strengthening tropical influence on heat generating circulation over
# Australia through spring

3          Roseanna C. McKay[1,2,4], Julie M. Arblaster[1,2,3], Pandora Hope[4]

[1] School of Earth, Atmosphere and Environment, Monash University, Victoria, Australia
[2] Australian Research Council Centre of Excellence for Climate Extremes
[3] National Center for Atmospheric Research, Boulder, Colorado, U.S.A.
[4] Bureau of Meteorology, Melbourne, Victoria, Australia
*Correspondence to:* Roseanna C. McKay (roseanna.mckay@bom.gov.au
## Abstract
Extreme maximum temperatures during Australian spring can have deleterious impacts on a
range of sectors from health to wine grapes to planning for wildfires, but are relatively
understudied compared to spring rainfall. Spring maximum temperatures in Australia have
been rising over recent decades, and, as such, it is important to understand how Australian
spring maximum temperatures develop. Australia's climate is influenced by variability in the
tropics and extratropics, but some of this influence impacts Australia differently from winter
to summer, and, consequently, may have different impacts on Australia as spring evolves.
Using linear regression analysis, this paper explores the atmospheric dynamics and remote
drivers of high maximum temperatures over the individual months of spring. We find that
the drivers of early spring maximum temperatures in Australia are more closely related to
low-level wind changes, which in turn are more related to the Southern Annular Mode than
variability in the tropics. By late spring, Australia's maximum temperatures are
proportionally more related to warming through subsidence than low-level wind changes,
and more closely related to tropical variability. This increased relationship with the tropical
variability is linked with the breakdown of the subtropical jet through spring and an
associated change in tropically-forced Rossby wave teleconnections. However, much of the
maximum temperature variability cannot be explained by either tropical or extratropical
variability. An improved understanding of how the extratropics and tropics projects onto the
mechanisms that drive high maximum temperatures through spring may lead to improved
sub-seasonal prediction of high temperatures in the future.



## 1. Introduction

Anomalously high Australian spring (September-October-November) maximum temperatures can be highly impactful. High temperatures may negatively impact health due to a lack of acclimatisation (e.g. Nairn and Fawcett, 2014), and agriculture by changing growing season length and crop yields (Cullen et al., 2009; Jarvis et al., 2019; Taylor et al., 2018). Hotter and drier spring conditions have been linked to an earlier start to (Dowdy, 2018) and preconditioning of (Abram et al., 2021) the summer fire season. The trend toward higher temperatures over recent decades (Collins et al., 2013), means that anomalous high maximum temperatures may occur more often (e.g. Alexander and Arblaster, 2009). Several recent springs have already exceeded historic temperature records, with some spring months breaking records that were set only the previous year (Arblaster et al., 2014; Gallant and Lewis, 2016; Hope et al., 2015; McKay et al., 2021). Much of this observed anomalous heat has been attributed to the background global warming trend (Arblaster et al., 2014; Gallant and Lewis, 2016; Hope et al., 2015; Hope et al. 2016). However, gaps remain in our understanding of what drives anomalous high maximum temperatures in Australia during spring, and particularly on the monthly time-scale that some of these heat events occurred over. As the globe continues to warm, a better understanding of what makes a spring month in Australia hot today will lead to greater resilience against extreme heat in the future.

High spring temperatures have been linked with several remote modes of variability in the tropics and extratropics. In the tropics, the positive phases of El Niño Southern Oscillation (ENSO) in the tropical Pacific and the Indian Ocean Dipole (IOD) in the tropical Indian Ocean are the strongest drivers of high maximum temperatures in Australia in spring, particularly in the south and east (Power et al., 1998; Jones and Trewin, 2000; Saji et al., 2005; Min et al., 2013; White et al., 2014). Many more studies focus on the ENSO and IOD relationships to drier spring conditions (Nicholls et al., 1989; Meyers et al., 2007; Ummenhofer et al., 2009; Risbey et al., 2009a; Watterson, 2010; Cai et al., 2011; Min et al., 2013; Pepler et al., 2014; McIntosh and Hendon, 2018; Watterson, 2020) and to more extreme spring fire weather (Harris and Lucas 2019; Marshall et al. 2021). While ENSO and the IOD co-vary significantly in austral spring (e.g. Meyers et al., 2007), they can occur independently (e.g.





Risbey et al 2009a). Further, the IOD's influence on Australia's temperature peaks around
SON (Saji et al., 2005) compared to around NDJ (November-December-January) for ENSO
(Jones and Trewin, 2000). It can be useful to look at a single index that describes the large-
scale tropical SST variability's influence on Australia, such as the tropical tripole index (TPI)
(Timbal and Hendon, 2011). While other tropical modes of variability, such as the Madden-
Julian Oscillation (MJO), also influence Australia's spring maximum temperatures (e.g.
Wheeler and Hendon, 2004; Wheeler et al., 2009; Marshall et al., 2014), we focus on the
tropical SST-driven influence on Australia's spring climate.

Variability in the extratropics is also linked to high temperatures in Australia. The negative
phase of the Southern Annular Mode (SAM), the primary mode of variability in the
extratropics, (Hendon 2007; Risbey et al., 2009a; Min et al., 2013; Marshall et al., 2012,
Hendon et al., 2014; Fogt and Marshall, 2020) drives hotter and drier Australian spring
conditions, and to more extreme spring fire weather (Marshall et al. 2021). SAM generally
varies at a higher frequency than ENSO or the IOD, however, SAM also has lower frequency
variations. On a seasonal timescale, El Niño promotes negative SAM, particularly during the
warmer months (L'Heureux and Thompson, 2006; Hendon et al., 2007; Lim et al., 2016; Lim
et al., 2019a). Polar stratospheric weakening during austral spring (sometimes associated
with sudden stratospheric warming) can also sustain negative SAM (Lim et al., 2018)) and
higher Australian maximum temperatures from late spring (Lim et al., 2019b). As with ENSO
and the IOD, more studies focus on how SAM influences Australian rainfall than
temperature, particularly when examining the teleconnection pathway. While low rainfall
correlates well with high maximum temperatures (Simmonds, 1998; Jones and Trewin,
2000; Timbal et al., 2002; Hope and Watterson, 2018), there is a gap in our understanding of
how both tropical and extratropical modes of variability impact spring maximum
temperature.

Anomalously high geopotential height (or, synonymously, anticyclonic vorticity) over
southern Australia is associated with spring high maximum temperatures in Australia (Hope
et al., 2015; Gallant and Lewis 2016; McKay et al., 2021). While ENSO, the IOD, and the
tropical TPI also promote anomalously high geopotential height, it forms further to the
south of Australia (e.g. Cai et al., 2011; Timbal and Hendon, 2011; McIntosh and Hendon,



2018). SAM's negative phase is characterised by an equatorward shift of the eddy-driven jet
and bands of anomalously low and high geopotential height in the mid- and high-latitudes
respectively (Thompson and Wallace, 2000; Fogt and Marshall, 2020). The altered
atmospheric flow associated with the drivers can reduce rainfall, including by defecting
cooling rain-bearing systems (e.g. Jones and Trewin 2000; Hendon et al., 2007; Pepler et al.,
2014; van Rensch et al., 2019) away from Australia (Cai et al., 2011; Risbey et al., 2009b;
McIntosh and Hendon, 2018; Hauser et al., 2020). Anomalous heat and dry is also
associated with other mechanisms such as increased subsidence and insolation (Hendon et
al., 2014; Lim et al., 2019b; Pfahl et al., 2015; Quinting and Reeder, 2017; Suarez-Gutierrez
et al., 2020) or heat advection (Jones and Trewin, 2000; Boschat et al., 2015; Gibson et al.,
2017). Understanding the differences between the extratropical and tropical forcing behind
some of these heat mechanisms is a goal of this paper.

The mechanisms and atmospheric circulation patterns associated with heat and connections
to remote drivers may also vary through spring. McKay et al. (2021) noted that the
relationship with the southern Australian upper-anticyclone and maximum temperature is
weaker in September than November, and suggested that the anticyclone had greater
influence from the tropics in later spring. The impact of SAM in the extratropics on
Australia's temperature reverses from winter to spring (Hendon et al., 2007; Risbey et al.,
2009a; Marshall et al., 2012; Min et al., 2013; Hendon et al., 2014; Fogt et al., 2020) as the
mean zonal winds change with the seasons (Hendon et al. 2007) and the Indo-Pacific
subtropical jet (STJ) weakens (Bals-Elsholz et al., 2001; Koch et al., 2006; Ceppi and
Hartmann, 2013, Gillett et al., 2021) so that a negative SAM phase enhances subsidence
over subtropical Australia into the warmer months (Hendon et al., 2014). The IOD and ENSO
teleconnection pathways over the Indian Ocean toward Australia also change from winter
to spring (Cai et al., 2011). This change may relate to the strength of the winter STJ, as it
should prevent direct propagation of Rossby waves between the tropics and extratropics
(e.g. Hoskins and Ambrizzi, 1993). McIntosh and Hendon (2018) proposed that transient
eddy-feedbacks generate a secondary wave source south of the winter STJ in response to
IOD forcing. In spring, the STJ weakens sufficiently to allow for direct Rossby wave
propagation from the tropical Indian Ocean. However, McKay et al. (2021) suggested that



the STJ may not weaken sufficiently in September to allow direct Rossby wave propagation,
and that teleconnection pathways may be different on a monthly timescale as a result.

Teleconnections driven by large-scale remote modes of variability can precondition
Australia toward hotter and spring conditions (e.g. Hurrell et al., 2009), but cannot
guarantee a hot month or season will eventuate. Even the strongest El Niño events may not
result in the canonical dry and warm conditions expected (van Rensch et al., 2019; Hauser et
al., 2020). Further, differences in how those modes of variability influence Australia
between winter-spring-summer and the differences between spring-average atmospheric
circulation highlight that there is more to understand in how maximum temperatures evolve
through spring months. Filling the gap between weather and seasonal time-scales is an
ongoing area of research that can lead to improved sub-seasonal forecasting (Meehl et al.,
2021). Given the increasing likelihood of future extreme heat events occurring through
spring, it is imperative to understand any differences that may exist in how heat develops,
and links to varying influences from the extratropics to tropics. The reanalysis datasets,
Rossby wave and statistical analysis methods are described in Section 2. An overview of how
Australian spring maximum temperatures are related to circulation and large-scale
variability is in Section 3. In Section 4 the variation of these relationships through the
months of spring are assessed and Section 5 describes how the drivers influence the
mechanisms that promote high monthly maximum temperature. Discussion and conclusions
are provided in Section 6

## 147    2. Methods and data

### 148    2.1 Indices and datasets

All circulation variables for September, October, November monthly-averaged data are
taken from the ECMWF's Reanalysis 5 (ERA5) (Hersbach et al., 2020) available from the
Copernicus Climate Change Service (C3S, 2017) on a 0.25° grid from 1979 to 2019. Low-level
circulation is diagnosed using 850hPa horizontal wind and mean sea level pressure (MSLP).
Mid-tropospheric vertical motion is represented by 500hPa omega. Upper-level circulation
is represented by 200hPa geopotential height (200Z). 200hPa horizontal winds are used for





Rossby wave analysis. Similar results were found using ERA-Interim reanalysis (Dee et al,
2011) and the JRA-55 from the Japan Meteorological Agency (2013) (not shown).

Australian monthly-averaged daily maximum temperature data for 1979 to 2019 is taken
from the Australian Water Availability Project (AWAP) (Jones et al., 2009) analyses, available
on a 0.05° resolution grid.

Monthly sea surface temperature (SST) is taken from NOAA Extended Reconstructed Sea
Surface Temperature (ERRSST V5; Huang et al., 2017)

The impacts of SAM on Australia's climate shows some sensitivity to the method used to
calculate the SAM index (e.g. Risbey et al., 2009a). To ensure consistency between the other
indices and circulation variables, we calculate SAM as the difference between the
standardized zonal means of ERA5 MSLP anomalies at 60°S and 40°S (Gong and Wang,

169    1999).


The tropical TPI (Timbal and Hendon, 2011) is defined as the difference in SST averaged over
a parallelogram located over the Maritime Continent (0°-20S, 90°-140E at the equator
shifted to 110°-160°E at 20°S) from SST averaged and summed over two regions in the
tropical Indian Ocean (10°N to 20°S, 55° to 90°E) and tropical Pacific Ocean (a trapezium
that extends from 15°N to 15°S, 150°E to 140°W in the north and 180°E to 140°W in the
south). ENSO is described using the Niño3.4 index (averaged SST anomalies over 5°N-5°S,
170°E-120°W) and the IOD using the dipole mode index (DMI; the difference between the
SST anomalies averaged over 10°S-10°N, 50°-70°E and 10°S-0°, 90°-110°E; Saji et al., 1999).

To highlight the influence of interannual variability, the 1981-2010 climatological mean is
removed from each month, and the data is linearly detrended before analysis.



2.3 Rossby wave analysis
We use wave activity flux (WAF) at 200hPa to trace Rossby wave group propagation and to
identify source and decay regions that influence the atmospheric circulation patterns.
Following Takaya and Nakamura (2001), we calculate WAF as:

$$WAF = p\cos\phi \left\{ \frac{U}{a^2\cos^2\phi} \left[ \left(\frac{\partial\psi'}{\partial\lambda}\right)^2 - \psi'\frac{\partial^2\psi'}{\partial\lambda^2} \right] \right.$$

$$+ \frac{V}{a^2\cos\phi} \left[ \frac{\partial\psi'}{\partial\lambda}\frac{\partial\psi'}{\partial\phi} - \psi'\frac{\partial^2\psi'}{\partial\lambda\partial\phi} \right] \quad \frac{U}{a^2\cos\phi} \left[ \frac{\partial\psi'}{\partial\lambda}\frac{\partial\psi'}{\partial\phi} - \psi'\frac{\partial^2\psi'}{\partial\lambda\partial\phi} \right]$$

$$\left. + \frac{V}{a^2} \left[ \left(\frac{\partial\psi'}{\partial\phi}\right)^2 - \psi'\frac{\partial^2\psi'}{\partial\phi^2} \right] \right\}$$


where $p$ is the pressure (200hPa) scaled against 1000hPa, $U$ and $V$ are the climatological
zonal and meridional wind speed magnitudes, a is the radius of the earth, $(\phi, \lambda)$ are latitude
and longitude, $\psi' = Z'/f$ is the quasi-geostrophic perturbation streamfunction, $Z'$ is the
200hPa geopotential height anomaly obtained through regression onto maximum
temperature or climate driver indices, $f = 2\Omega\sin\phi$ is the Coriolis parameter with the
Earth's rotation $\Omega$. WAF is not plotted within 10° of the equator.

WAF propagates in the direction of quasi-stationary Rossby wave group velocity, and
regions of divergence or convergence of WAF correspond to zones of Rossby wave sources
or sinks respectively.

Total stationary Rossby wave wavenumber (e.g., Hoskins and Karoly 1981) **is** defined as:
$$K_S = \sqrt{\frac{\beta - U_{yy}}{U}}$$

where $\beta - U_{yy}$ is the meridional gradient of mean-state absolute vorticity at 200hPa. WAF
should refract toward regions of higher $K_S$ and either reflect or evanesce on regions of $K_S<0$,
such as in the STJ where the curvature of the flow ($U_{yy}$) can become larger than the
planetary vorticity gradient ($\beta$) (e.g Barnes and Hartmann, 2012; Li et al., 2015 a,b)



### 2.4 Statistical analysis


Linear, partial, and multi-linear regression and Spearman's ranked correlation are used to
assess the relationships between Australian maximum temperature, atmospheric circulation
and the tropics and extratropics. Due to the large decorrelation length scales, Australian-
average maximum temperature variability is representative of all but far north Australia's
spring and spring-monthly maximum temperatures (Sup. Fig. 1). Statistical significance is
calculated at the 95% confidence level using Student's (1908) t-test using 39 (41 years - 2)
degrees of freedom. Pattern correlation is used to compare regression patterns.

## 3. Spring-season maximum temperatures - circulation patterns and associations with drivers




We start by giving an overview of the spring-seasonal relationships between average
Australian austral spring maximum temperature and lower- and upper-level atmospheric
circulation (Fig 1a,b). Barotropic cyclones appear to the southwest and southeast of
Australia, occurring in both the lower- and upper-level circulation regressions (Fig. 1a-b) and
noted during recent extreme spring heat events (Gallant and Lewis, 2016; Hope et al., 2016;
McKay et al., 2021). Weak anticyclonic low-level winds are found over Australia, as well as
sinking motion across the eastern half of the continent. An upper-level anticyclone sits over
southern Australia, with the wave activity flux predominantly propagating from the
subtropical Indian Ocean, through the anticyclone and into the subtropical Pacific Ocean.

We now compare the atmospheric patters associated with spring maximum temperature to
those associated with large scale modes of variability. The spring-average atmospheric
circulation patterns associated with the remote drivers of variability are calculated via linear
regression onto each standardised index. Note that the TPI and SAM indices have been
multiplied by negative one to present positive associations with high temperatures. The
pattern for SAM (x-1) shows elongated barotropic low and high anomalies lie in the middle
and high latitudes respectively (Fig. 1c-d), with upper-level cyclonic nodes to the southeast





and southwest of Australia. Negative SAM is associated with high maximum temperatures
through much of subtropical, and particularly eastern, Australia (Fig. 1e).

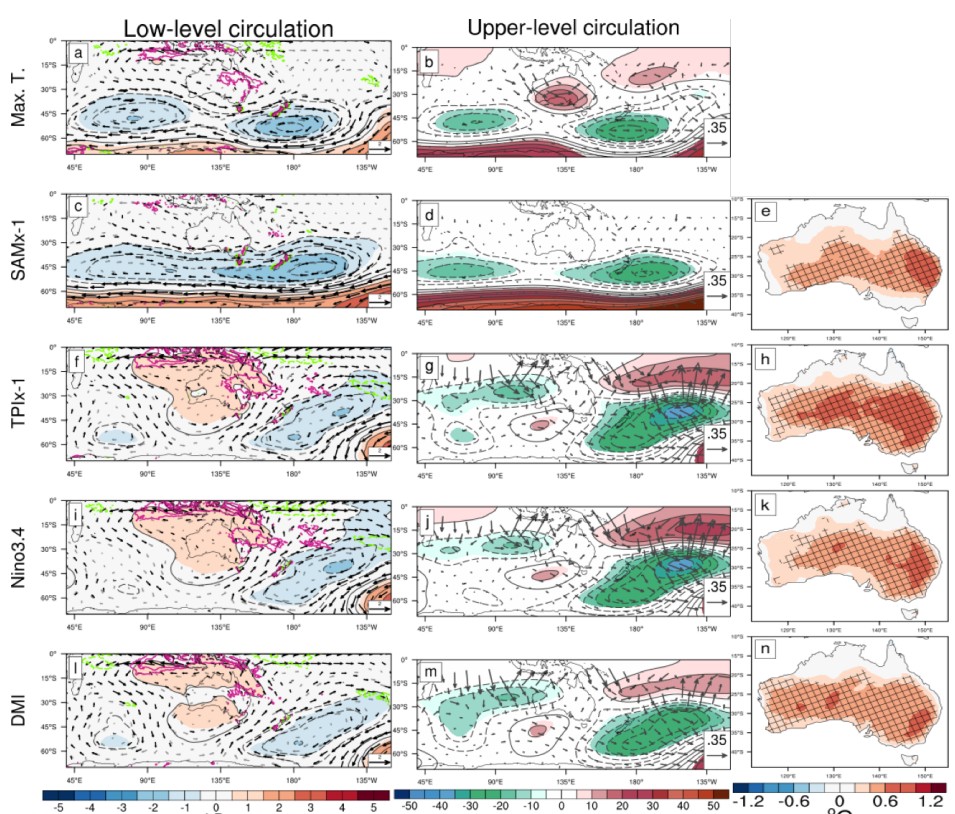

Figure 1. Linear regressions of spring standardised weighted area-averaged Australian maximum temperature (a-b), SAMx-1 (c-e), tropical TPIx-1 (f-h), Niño3.4 (i-j) and DMI (j-n) onto low-level circulation (left column), upper-level circulation (middle column) and Australian maximum temperatures (right column). Low-level circulation is represented by anomalous mean sea level pressure (hPa) (black and filled contours), 850hPa wind vectors (ms$^{-1}$) and 500hPa omega (hPas$^{-1}$) contours from -0.02 to 0.02hPas$^{-1}$ in steps of 0.01 hPas$^{-1}$ (magenta contours are positive; downward motion) and cyan contours are negative; upward motion, and the zero contour is not plotted). Upper-level circulation is represented by 200hPa geopotential height (black and filled contours and wave activity flux vectors (m$^2$s$^{-2}$).
Filled contours, bold wind vectors, cross-hatching, and all vertical motion contours are significant at the 95% confidence level using a Student's t-test with 39 independent


The tropical modes, represented by Niño3.4 (Fig 1i-k), the DMI (Fig. 1 l-n), and tropical TPI
(x-1) (Fig 1f-h)) are also associated with spring Australian maximum temperature anomalies.
Each mode generates an apparent Rossby wave pattern that arcs from the tropical Indian





Ocean to promote anomalous high geopotential height south of Australia, consistent with
earlier studies (e.g. Cai et al., 2011; Timbal and Hendon, 2011; McIntosh and Hendon, 2018).
Each regression also shares anomalous high surface pressure over Australia, sinking motion
in the east, cyclonic nodes to the southwest and east of Australia, and elongated upper-level
cyclones in the subtropical Indian Ocean. These similarities are likely the result of the strong
co-variability between the IOD and ENSO (e.g. Meyers et al., 2007; Risbey et al., 2009a).
However, the IOD has a stronger low-level cyclone to the southeast and a poleward
extension of the subtropical Indian Ocean cyclone that sets a subtly different wave train
from around 50°S, 60°E that is poleward that generated by ENSO. The positive IOD is also
associated with high maximum temperatures across a broader region of southern and
western Australian than is El Niño. The tropical TPI (x-1) is a blend of both Niño3.4 and DMI
circulation patterns and has a strong relationship with Australian spring maximum
temperatures across all but northern Australia.

Given the similarities and connections between ENSO and IOD teleconnections, we use the
tropical TPI to represent the large-scale influence of the tropics. SAM is used to represent
the influence of the extratropics. Statistical models of Australian weighted area-averaged
spring maximum temperatures reconstructed through multilinear regression using either
Niño3.4, DMI and, SAM or the tropical TPI and SAM as the predictors explains 32% and 34%
of maximum temperature variability respectively (sup. Fig. 2).

We next compare the atmospheric circulation associated with monthly high maximum
temperatures to that with the large-scale modes of variability through the individual months
of spring. To ensure that we are assessing the influence of the tropics and extratropics
separately, we use multi-linear regression onto the monthly circulation variables.

## 272 4. Monthly circulation patterns and associations with drivers

The regression of monthly Australian maximum temperature onto the lower- and upper-
level atmospheric circulation is displayed in Figures 2a-c and 3a-c respectively for
September, October and November. The multi-linear regression onto the standardised
monthly indices of SAM (x-1) (Figs. 2d-f and 3d-f) and tropical TPI (x-1) (Figs. 2h-j and 3h-j).




At first glance, these monthly circulation patterns are broadly similar to the spring-average
regression patterns. However, the details of the circulation patterns change as the months
progress, suggesting that different processes are important for heat development through
spring.

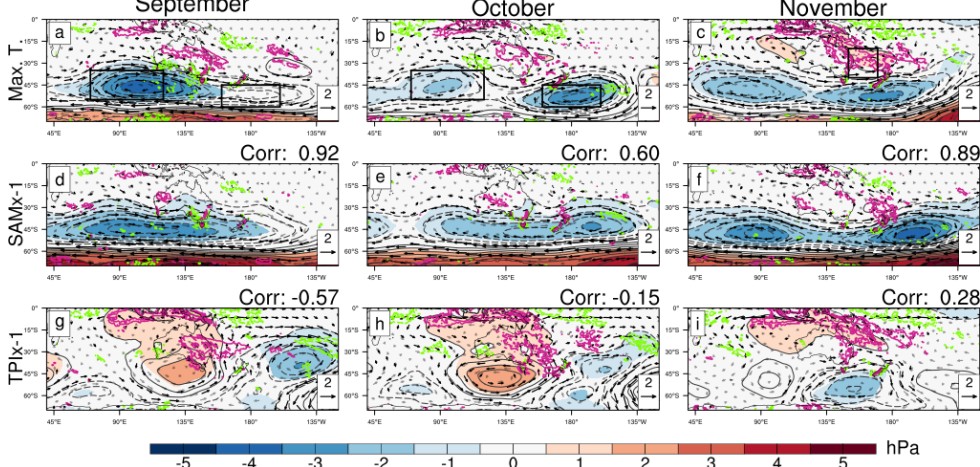

Figure 2. Regressions onto low-level circulation, as in Fig. 1, except for September (left column), October (middle column) and November (right column). Standardised areal-averaged Australian maximum temperature is linearly regressed onto low-level circulation (ta-c) and SAMx-1 (d-f) and tropical TPIx-1 (g-i) are multi-linear regressions onto low-level circulation.
Pattern correlation between the maximum temperature MSLP regressions and the SAM and tropical TPI regressions calculated over 5°S-70°S; 70°E-170°E are written in the top right of each SAM or tropical TPI regression.
The boxes (a-c) show key low-level circulation features identified as being important for maximum temperature development: The southwest cyclone (SWC) 35°S-55°S; 70°-120°E; southeast cyclone (SEC) 45°S-60°S; 160°-200°E and Tasman Sea high (TSH) 20°S-40°S; 150°-170°E


The most obvious change in atmospheric circulation through the months is in the low-level
flow across Australia, particularly generated by the barotropic cyclones southwest (SWC) or
southeast (SEC) of Australia (boxes in Fig 2a-b). Weak low-level anticyclonic flow around the
Tasman Sea (box in Fig. 2c) also contributes to the northerly flow over eastern Australia in
November in particular (Fig. 2c). Tasman Sea anticyclonic blocking patterns have previously



been linked to anomalously warm conditions (Marshall et al., 2014), but here appear to only
contribute to high maximum temperatures in November. The SWC and SEC vary in
geographic shape and strength through the months. The SWC dominates in September but
weakens through October and November, whereas, the SEC is missing in September but is
strong in October and November. Similar cyclones appear in the monthly SAM (x-1)
regressions (Figs. 2 d-f, 3d-f) and the Australian-region MSLP correlates strongly with that
associated with high Australian temperature (top-right of Fig 2d-f). Rather than cyclones in
September and October the TPI (x-1) is associated with a barotropic anticyclone south of
Australia that directs southerly low-level wind across eastern Australia (Figs. 2 h-j); a pattern
that would be associated with cooler conditions. The September and October TPI (x-1) MSLP
pattern actually anti-correlate with that associated with high Australian maximum
temperatures (top-right Fig. 2 g-i). It is not until November that we see a barotropic cyclone
to the southeast of Australia associated with the TPI (x-1). So, for the majority of spring
negative TPI-forced low-level atmospheric circulation appears to counter high maximum
temperatures, despite the overall positive relationship in spring (Fig. 1h).

The anomalous southern Australian upper-anticyclone (SAA) from the spring pattern
appears is also associated with high maximum temperature in each of the individual spring
months (Fig. 3a-c), but its location shifts eastward across Australia through spring. The
boxed region was chosen to match earlier studies (Gallant and Lewis, 2016; McKay et al.,
2021), but best matches the November position, likely contributing to the stronger
relationship between heat and SAA in this month (McKay et al., 2021; see also section 6).
The anticyclone in later spring appears to form part of a wave train from a cyclone to the
northwest of Australia toward the southeast cyclone. While the monthly TPI regressions
have anticyclones in September and October (Fig. 3g-h), they are located too far south
relative to Australia, as in the spring-average regression. The regressions onto SAM (x-1) (Fig
3d-f) have weak anticyclones over western Australia that are not statistically significant. It is
not until November that both SAM and TPI (x-1) (Figs 3 f, i) have an anticyclone over central-
east southern Australia. Both the upper- level SAM and TPI (x-1) regressions correlate
moderately with the maximum temperature regression in November, and the SAM and TPI
(x-1) anticyclones may form part of the same wave train associated with maximum
temperatures. However, the SAM and TPI (x-1) anticyclones are weaker and too far east



relative that associated with high maximum temperatures, such that they may not
contribute strongly to the SAA formation. We explore this idea further in section 6.

Figure 3. As with figure 2, but for upper-level circulation. The Australian-region pattern
correlation between the maximum temperature Z200 regressions and SAMx-1 and TPIx-1
are in the top right of each figure. Boxed area (a-c) highlights the southern Australian
anticyclone (SAA; 30°-40°S, 120°-150°E) that is linked with high maximum temperatures.


While the southern Australian anticyclone is not well explained by SAM or TPI (x-1) through
spring, much of the statistically significant 500hPa vertical motion associated high maximum
temperatures (green and magenta contours, Fig. 2a-c) matches that associated with TPI (Fig.
2h-j) and to a lesser extent SAM (Fig. 2d-f). In September, sinking motion over subtropical
Australia and rising motion over the southern coasts is associated with high maximum
temperatures. By November, the rising motion has largely vanished and the sinking motion
has shifted to be over eastern Australia. It was expected that the SAA would generate some
of the sinking motion associated with high maximum temperature, however, this vertical
motion does not correlate strongly with any of the key circulation features examined here
(Sup. Table1).



Changes in propagation of wave activity flux help explain some of the changes in the broad
scale circulation changes through spring. In September, WAF predominantly propagates
from the southwest cyclone toward the southern Australian anticyclone. In October, a
component of WAF also propagates from the eastern tropical Indian Ocean region. By
November, the tropical-component dominates the WAF and forms part of a very different
pattern to the previous two months; continuous WAF propagates from the far southwest
Indian Ocean, joins WAF propagating out of the tropical Indian Ocean and then continues
across the southern Australian anticyclone. The latter part of this wave train is similar to the
IOD teleconnection highlighted by Cai et al. (2011) in spring. The WAF associated with SAM
and TPI (x-1) also propagates from the extratropics toward the respective anticyclones in
September and October. While a broad region of low height in the subtropical Indian Ocean
is associated with the TPI, it does not appear to generate WAF that propagates into the
extratropics. It is not until November that WAF associated with the TPI (x-1), and weakly
with SAM (x-1), appears to propagate directly from the cyclone in the eastern subtropical
Indian Ocean through the anticyclone over southeastern Australia.

Overall, these results suggest that the circulation associated with maximum temperature
shifts from extratropical to tropical forcing as spring progresses. This is supported by how
well SAM appears to project onto the atmospheric circulation associated with maximum
temperatures in September, and how the TPI projects more strongly later in spring. The
change in WAF associated with this change suggests that there may be a blocking
mechanism between the tropics and extratropics generating this change.

We find qualitatively similar results if we perform the linear regressions using maximum
temperature averaged over sub-regions of Australia, for example southwest or southeast
Australia (Supplemental Fig S2).

## 362    5. Connection between subtropical jet and atmospheric circulation

We next explore how the subtropical jet may be influencing the WAF through the spring
months.

The subtropical jet (STJ) peaks in strength in winter and weakens through spring to have
broken down by summer (e.g. see figure 9, Ceppi and Hartmann, 2013). This gradual
breakdown of the STJ coincides with a decrease in the area with total stationary
wavenumber less than zero over southern Australia (Fig. 4), and may provide an explanation
for the growing relationship with the tropics and Australian maximum temperature by
November.

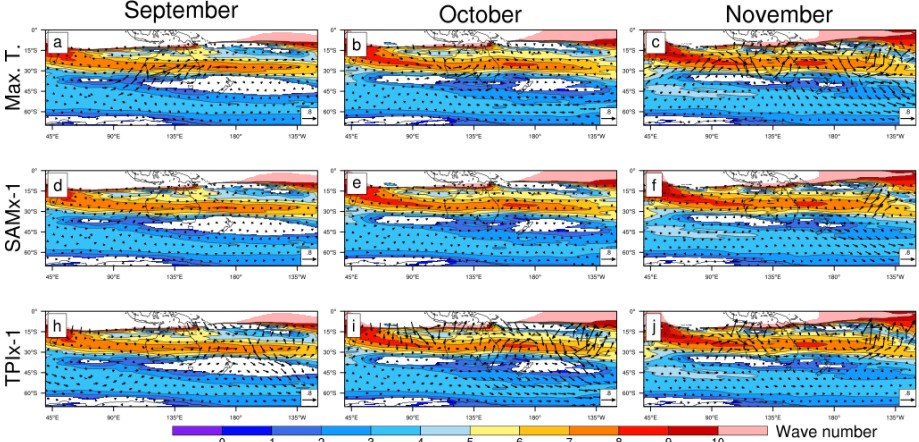

Figure 4. Total wave number (K) calculated for September, October and November.
Vectors are the wave activity flux repeated from figure 3.


The wave activity flux vectors from the maximum temperature, TPI and SAM (x-1)
regressions in figure 3 are overlaid in figure 4 on the monthly climatological Ks associated
with the zonal winds. In September, the WAF associated with high maximum temperature
(Fig. 4a) diverges from the region of the southwest cyclone to propagate through a region of
low total stationary Rossby wave wavenumber, Ks, over southwest Australia and along the
STJ waveguide (i.e. from high to low latitudes). As the jet weakens in October (Fig. 4b) a
portion of WAF also diverges from the tropical Indian Ocean to dissipate on the jet's
equatorward flank, but mostly propagates from west to east along the STJ waveguide. Even
more distinctive, by November (Fig 4c), WAF propagates along the jet waveguide to a region
near Africa, with contributions from the tropical Indian Ocean, but does not appear to
propagate out of the SWC.

The increase in WAF associated with the tropical TPI (Fig 4h-j) propagating out of the
tropical Indian Ocean through spring appears to coincide with the STJ decay. In September



and October weak WAF diverging from the central southern Indian Ocean follows the eddy-
driven jet waveguide (region of locally higher wave number around 50°S), suggesting the
secondary wave source proposed by McIntosh and Hendon (2018) is important in early
spring. The tendency for TPI-associated WAF to form and follow this trajectory may explain
why the barotropic anticyclone associated with the TPI is further poleward than in the
regression onto Australian maximum temperature. By October more WAF is propagating
out of the tropical Indian Ocean along the region of high Ks and by November WAF is
propagating out of the extratropical Indian Ocean along the high Ks region, similar to the
maximum temperature-WAF. WAF generated by SAM (Fig. 4d-f) also converges toward the
STJ waveguide in each month.

Limits around linear Rossby wave theory (e.g. Liu & Alexander, 2007) may explain why some
wave activity flux cross the region of imaginary wavenumber associated with the STJ.
However, the majority of WAF associated with Australian maximum temperature, or with
the tropics or extratropics does divert to propagate along the jet, as expected. While the
breakdown of the STJ through spring may help explain the change in teleconnection
pathways of the TPI toward Australia, the STJ consistently acts as a waveguide toward
Australia.

We now look more closely into how the drivers, circulation features, and heat mechanisms
relate to each other and how that results in higher Australian maximum temperatures.

## 6. Mechanisms and drivers of monthly maximum temperatures through spring
As with the atmospheric circulation regressions, the relationships between Australian
maximum temperature and SAM and TPI (x-1) evolve through the spring months. In
September, negative SAM (Fig. 5a) is associated with a broad area of high maximum
temperature over subtropical Australia, that contracts in October and November (Figs. 5 b-
c). Conversely, the relationship with negative TPI and maximum temperature is weaker early
in spring, with statistically significant high temperatures confined to the west and east, and
cool temperatures in the far north in September (Fig. 5d). The TPI's relationship with high
maximum temperature broadens and strengthens in October and covers the majority of



Australia by November (Figs. 5 e-f). Overall, these monthly relationships give the impression
of a transition from extratropical to tropical drivers becoming more influential over
Australian temperatures that is broadly consistent with the apparent change in atmospheric
circulation through spring.

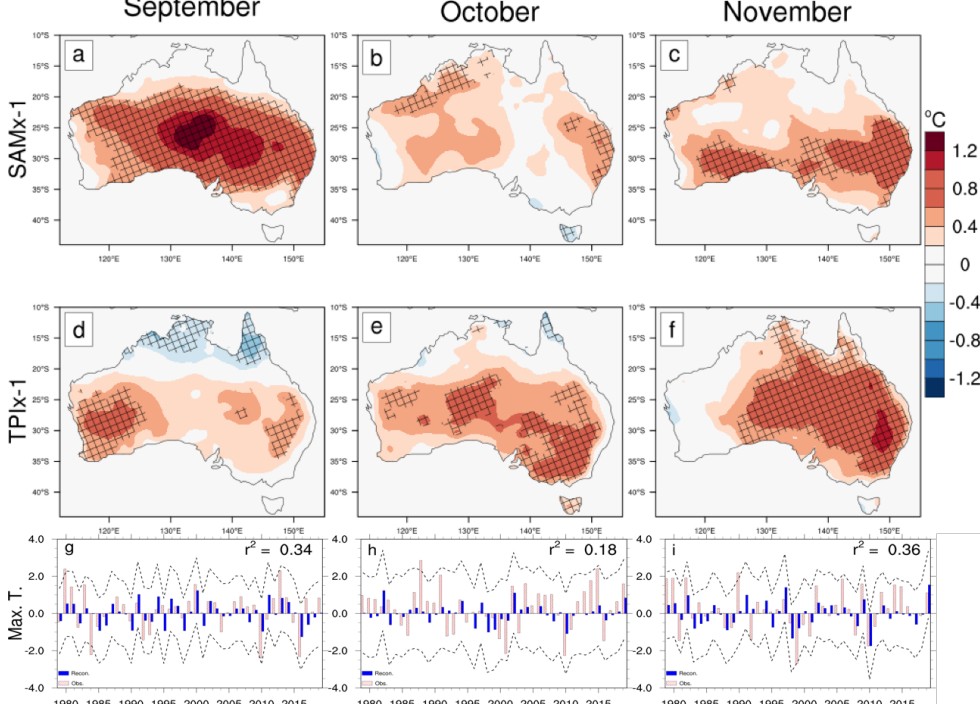

Figure 5. Multilinear regression coefficients (°C) of Australian maximum temperature regressed onto standardised timeseries of the SAMx-1 (a –c ) and the tropical TPI x-1 (d - f) for September, October and November over the years 1979 to 2019. Reconstructions (blue bars) of September, October and November (i-k) Australian area-averaged maximum temperature from standardised time series of SAM and tropical TPI indices. Observed values are in blue. The dashed line shows the 95% prediction interval computed as +/-1.96 standard error and the variance explained ($r^2$) of the model is in the top right of each figure.


Using the standardised SAM and TPI time series as predictors in a regression model to
reconstruct the monthly Australian-averaged maximum temperature anomalies (Figs 4g-i)
explains only between 18 and 36% Australian maximum temperature variance ($r^2$) through
spring. The model does not substantially improve if it is calculated over southeast or
southwest Australia, or if using Niño3.4 or DMI as predictors instead of the tropical TPI (Sup.
Fig.4).






To explore how the atmospheric circulation relates to some of the mechanisms that develop
heat through spring, we first compose indices of the key circulation features discussed in
section 4. Weighted area-averages of mean-sea level pressure (multiplied by negative one)
over the southwest and southeast cyclones (SWC and SEC) and 200hPa geopotential height
over the southern Australian anticyclone (SAA) for each spring month. See Figs. 2a,b and 3a-
c for regions. Creating a statistical model of Australian-averaged monthly spring maximum
temperatures from these circulation features (Fig. 6a-c) explains consistently higher
maximum temperature variance (around 60%) than did the model from the indices of
tropical and extratropical large-scale modes of variability. Further, despite the changes in
the features' geographic shape, strength and position across the spring months in Fig .2, the
majority of maximum temperature across Australia is well explained by at least one of these
features at all times through spring (Sup. Fig. 6). We next explore how these MSLP or
200hPa geopotential height features relate to the low-level westerly or northerly winds and
vertical motion and how that relates to high maximum temperature development.

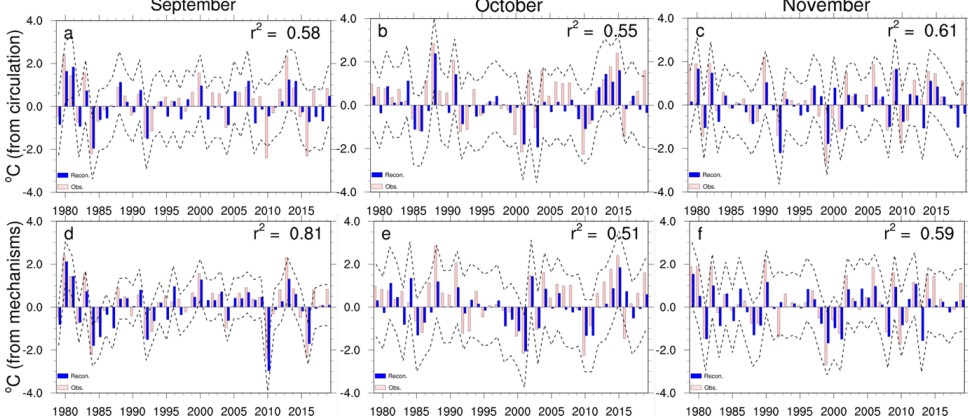

Figure 6. As in figure 4 (g-i), but using time series of key circulation features (south-west low, south-east low and southern Australian anticyclone) identified in figures 1 and 2 as predictors in the top row (a-c) and area-averaged dynamical heat mechanism components (850hPa zonal wind and meridional wind (multiplied by -1) and 500hPa vertical motion; see text for region averaged over) as predictors in the bottom row (d-e) for September, October, and November.


Following van Rensch et al (2019), indices of three dynamical heat mechanisms were
created by weighted area-averaging of westerly and northerly wind (meridional wind
multiplied by -1) over a region around southern Australia (25°S-45°S, 105°-155°E), and





500hPa vertical motion (omega; positive is sinking motion) averaged over subtropical
Australia (15°S-25°S, 120°-155°E). Regions were selected based on the areas of highest
statistical significance between atmospheric circulation and Australian maximum
temperature in Fig. 2a-c. Again, a statistical model of Australian-averaged maximum
temperatures that uses these mechanisms as the predictors explains a higher proportion of
maximum temperature variance through spring than does the model using SAM or the
tropical TPI (Fig. 6 d-e). The percent variance explained is much higher in September (about
80%), before dropping to around 55% in October-November. The decrease in the percent
variance explained appears to be primarily associated with how strongly the westerly winds
correlate with maximum temperature over southern Australia; strong positive relationship
with westerly wind in September changes to insignificant or negative in October and
November (Supp. Fig S7 a-c). There is also an increase in negative correlation between
maximum temperatures and northerly winds in north-eastern Australia (Supp. Fig S7 d-e)
that will partly offset the increasing positive relationship further south. These changing
relationships between dynamical mechanisms and maximum temperature through spring
are linked with the changing relationships with the circulation features (Supp. Table 1)
through spring. Overall, however, the three dynamical heat mechanisms explain much of
Australia's monthly spring maximum temperature variability.

Figure 7 summarises the relationship between Australian maximum temperatures,
circulation features, dynamical heat mechanisms and climate drivers through the spring
months. The correlation between the SEC and Australian maximum temperature is
strongest in September and rapidly decreases through October and November, while
simultaneously the correlations with the SWC and particularly the SAA increase. As expected
from Fig. 2, the SEC and SWC are more closely linked with the extratropics. Linearly
regressing out the SAM component from time series of the SWC and SEC reduces the
correlation strength with Australian maximum temperature (Fig. 7a), particularly in
September. Conversely, linearly removing the tropical TPI slightly increases the correlation
between the cyclones and temperature, with the partial-correlation only weakening in
November. As SAM is strongly related to the barotropic cyclones it is also strongly related to
how temperature changes with the westerly wind. Linearly removing SAM from the
westerly wind time series nearly halves the correlation with maximum temperature in



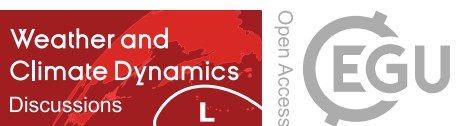

September, and weakens the correlation in October and November (Fig. 7b). Conversely,
linearly removing the tropical TPI actually increases the correlation slightly with the westerly
wind in September and October, but decreases the correlation in November.


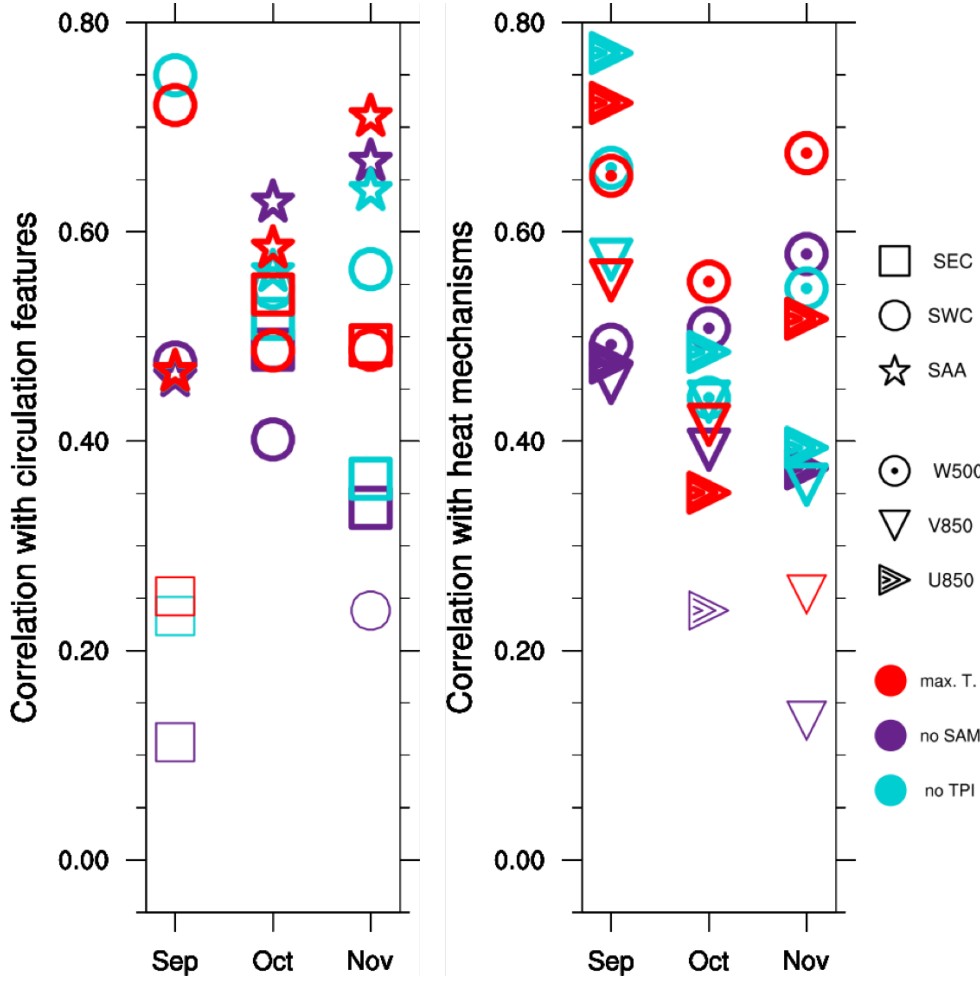

Figure 7. Correlations between Australian area-averaged maximum temperature (red) between key atmospheric circulation features (left figure) and dynamical heat mechanisms (right figure) for September, October and November. The purple and turquoise show partial correlations of the same, but with SAM and the tropical TPI linearly removed. Bold lines show the correlation was statistically significant at the 95% confidence level using a Student's t-test with 39 samples.



The relationships with northerly wind and sinking motion and Australian-averaged
maximum temperature do not change as dramatically with the removal of SAM or the TPI.
Northerly wind is not strongly influenced by the tropics or extratropics in September or
October, but the correlation strengthens and weakens in November with the removal of the
TPI and SAM, respectively. While removing SAM and TPI from the SAA had relatively little
influence on the correlation with Australian maximum temperatures, removing SAM from
sinking motion in September and both TPI and SAM in October and November reduced the
correlation. Overall, it appears that the heat mechanisms associated with high maximum
temperatures in spring are influenced differently by the different influence of the
extratropics and tropics on the local atmospheric circulation features through spring.

## 7. Discussion and conclusions
The sources of the atmospheric circulation pattern associated with high monthly-maximum
temperatures in Australia appear to change from primarily extratropical in early spring to
tropical forcing in late spring. Examination of three dynamical heat mechanisms (low-level
winds broken into westerly and northerly components, and mid-tropospheric sinking
motion) indicates that this shift may be due to a change in how heat develops. In early
spring, the low-level wind plays a greater role in maximum temperatures, advecting
relatively warmer air from the oceans over the cold land-mass. This wind correlates strongly
with the extratropics (here, SAM) as SAM projects strongly onto the southwest and
southeast cyclones that direct a lot of the low-level flow around Australia. Conversely, the
atmospheric circulation associated with the TPI (x-1) acts to counter the low-level flow that
drives higher temperatures. Thus, in early spring we have a closer association with heat
production and the extratropics. By late spring, the circulation patterns associated with high
temperature have changed and the wind does not correlate as strongly. As such adiabatic
sinking over subtropical Australia has a proportionally stronger correlation with high
temperatures. Both SAM and TPI (x-1) regressions show sinking motion in the subtropics
through spring, but it is the TPI that better matches the sinking motion over eastern
Australia in November. Hence, the apparent change from extratropical to tropical forcing in
the circulation pattern is because the tropics promotes more of the heat developing





mechanisms later in spring. However, much of the atmospheric patterns associated with
heat through spring are explained by neither the tropical TPI nor SAM,

The subtropical jet appears to play a greater role in Australian spring heat by acting as a
wave guide (Hoskins and Ambrizzi, 1993) that directs quasi-stationary Rossby waves toward
Australia, rather than as a block that limits direct propagation of Rossby waves from the
tropical Indian Ocean to the southern hemisphere extratropics (e.g. Simpkins et al., 2014; Li
et al., 2015 a,b). While wave activity flux only appears to propagate directly out of the
tropical Indian Ocean later in spring, this analysis does not suggest that the tropical Indian
Ocean is not a wave source in early spring. Indeed, the results are broadly consistent with
IOD-forced wave trains identified in the literature (Cai et al., 2011; McIntosh and Hendon,
2018; Wang et al., 2019). In particular, the secondary wave source in the high latitudes of
the Indian Ocean proposed by McIntosh and Hendon (2018) may be key for promoting the
TPI-forced atmospheric circulation in early spring, though this is beyond the scope of this
study to confirm. As the subtropical jet did not act as a barrier preventing the tropical Indian
Ocean's influence on Australia's maximum temperature, we argue instead that the apparent
change in forcing through spring was more related to the origins of three of the dynamical
heat mechanisms behind that heat. Consistent with this idea, wave activity flux calculated
by first regressing 200Z onto the three dynamical heat mechanisms (Sup. Fig. 8) also has
changing extratropical or tropical forcing through spring, that then propagates along the jet
wave guide toward Australia.

Area-averaged low-level wind and vertical motion were used to understand how the
atmospheric circulation relates to Australia-wide maximum temperatures, but do not form a
complete picture of spring temperature development in Australia. Statistical models using
these mechanisms explain much, but not all, of the maximum temperature variance over
Australia. Further, it was not always clear how the atmospheric circulation features
influenced those heat mechanisms. In particular, the southern Australian anticyclone and
500hPa subtropical-Australian sinking motion, while important for heat, appear to be largely
uncorrelated with the other circulation features and mechanisms. Greater insight into how
remote forcing of the atmospheric circulation results in high Australian temperatures could
be gained by including other heat mechanisms in future analyses, including: insolation (Lim



et al., 2019b), land-surface feedbacks linked to antecedent moisture (e.g. Arblaster et al.,
2014; Hirsch and King, 2020), and changes to synoptic weather systems (Cai et al., 2011;
Hauser et al., 2020). How each of these mechanisms relates to the others, and geographic
changes across Australia should also be considered. The combination of poleward advection
of adiabatically warmed air after it descended anticyclonically over the Tasman Sea has
been identified as a key mechanism for summer heatwaves in southeast Australia (e.g.
Quinting and Reeder, 2017). This combination of mechanisms may generate heat through
spring, particularly in the east and in November. The connection with rising motion over
southern Australia has also not been examined, and may indicate the importance of air
being diabatically warmed in association with storminess just to Australia's south, before
advecting and descending toward Australia. While the three heat dynamical heat
mechanisms were simple, the complex relationships between all of the mechanisms meant
that the three used in this analysis were broadly representative of a large portion of how
heat develops through spring.

We used the TPI to represent tropical variability relevant to Australia's maximum
temperature, but other indices or drivers may highlight different Rossby wave pathways or
heat mechanisms. Reconstructing Australian maximum temperature time series with more
commonly used indices for the IOD and ENSO did not change the effectiveness of the
statistical models overall (Supp. Fig. 4). However, it did suggest that the IOD had greater
influence on Australia's maximum temperature in early spring than does ENSO, consistent
with the seasonal-length studies of (Jones and Trewin, 2000; Saji et al., 2005). As such, we
may expect different monthly Rossby wave pathways to Australia associate with the IOD in
early spring, giving greater influence from the tropical Indian Ocean at this time. The MJO
generates Rossby wave trains from the western Pacific that promote low minimum
temperatures in Australia during winter (Wang and Hendon, 2020) and from the tropical
Indian Ocean to promote high maximum temperatures in Australia in spring (personal
communication: Wang and Hendon, 2021). The positive phase of the IOD suppresses MJO
activity across the Indian Ocean (Wilson et al., 2013), possibly restricting the MJO's
influence on Australia's maximum temperature at such times. However, MJO activity in the
tropical Indian Ocean has recently been found to counter the wetting influence of La Niña
during spring (Lim et al., 2021b). As such the MJO may be an important factor for spring



maximum temperatures when the tropical SSTs are not otherwise conducive for high
temperatures, but is beyond the scope of this study.

As the trend toward higher Australian spring temperatures is projected to continue into the
future a better understanding of what drives maximum temperatures over the months of
spring is critical for better prediction and better preparation to adapt to a warming climate.
A combination of extreme values in remote drivers of variability, including extreme positive
IOD, central-Pacific El Niño, and sustained negative SAM associated with very strong sudden
stratospheric warming, exacerbated already dry and hot conditions in spring 2019 to
promote one of Australia's deadliest fire seasons (Watterson, 2020; Lim et al., 2021a;
Abram et al., 2021, Marshall et al. 2021). Further, projected trends toward positive IOD (Cai
et al., 2014; Abram et al., 2020) or toward negative TPI (Timbal and Hendon, 2011) may
contribute to higher maximum temperatures in the future, particularly in later spring when
the tropics exert greater influence on Australia's dynamical heat mechanisms. As we have
shown just how different the atmospheric circulation and heat mechanisms can be through
a season in Australia, other regions and seasons could also benefit from similar analysis,
particular as the world continues to warm (e.g. Collins, et al., 2013).

## Code and data availability

The code for analysis is available from the corresponding author on request. ERA5-
reanalysis data are available from Copernicus Climate Change Service at
https://www.ecmwf.int/en/forecasts/datasets/reanalysis-datasets/era5. AWAP data is
available from the Australian Bureau of Meteorology.

## Author contribution

R.M.C produced the figures and wrote the initial draft manuscript. All authors contributed
to analysis and editing of the manuscript.

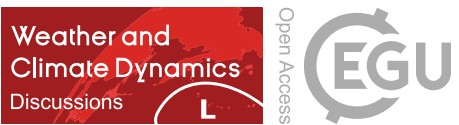

## Competing interests

The authors declare that there are no conflicts of interests.

## Acknowledgements

This research was supported by the Australian Research Council (ARC) Centre of Excellence
for Climate Extremes (CE170100023). R.M. was also supported by an Australian Government
Research Training Program (RTP) Scholarship and a Bureau of Meteorology PhD Top-up
scholarship. J.M.A. was partially supported by the Regional and Global Model Analysis
component of the Earth and Environmental System Modeling Program of the US
Department of Energy's Office of Biological & Environmental Research via National Science
Foundation IA 1947282. PH was supported by funding the Earth Systems and Climate
Change Hub of the Australian Government's National Environmental Science Program
(NESP). This research was undertaken at the NCI National Facility in Canberra, Australia,
which is supported by the Australian Commonwealth Government. The NCAR Command
Language (NCL; http://www.ncl.ucar.edu) version 6.4.0 was used for data analysis and
visualization of the results. We thank G. Wang for his assistance writing code for analysis, Z.
Gillett, E-P Lim, and H. Hendon for their insights into the data. The authors thank T.
Turkington for supplying the tropical tripole index dataset. We thank H. Hendon and A.
Marshall for their constructive feedback on the manuscript.

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
