# Peer review of "Strengthening tropical influence on heat generating circulation over"

_Weather and Climate Dynamics, 2021_

## Author Response (AR1)

Manuscript: WCD-2021-63

Strengthening tropical influence on heat generating circulation over Australia

through spring

by Roseanna C. McKay, Julie M. Arblaster, and Pandora Hope

Final author comments

We thank the two reviewers for their evaluation of our study and their comments. Their suggestions are very helpful and will greatly improve the manuscript. The reviewer's comments are in blue and our responses are in bold.

**Reviewer #1**

**Synopsis:**

This study of McKay et al. aims to quantify the effect of remote drivers on spring maximum temperatures in Australia. Focusing on ENSO, IOD, and SAM, the authors find that early spring maximum temperatures are more closely related to the extratropical circulation anomalies and late spring maximum temperatures are rather due to tropical variability. Analysis of the Rossby wave activity flux reveals that the increasing influence of the tropics is due to Rossby wave teleconnections emerging from the tropical Indian Ocean. Overall, the results are presented in a clear manner and the conclusions are justified. However, there are numerous instances where clarification is needed. In particular, the introduction could become more streamlined by restructuring parts of it (see major comment #1). Once the comments have been addressed, I think this paper is a worthwhile contribution to Weather and Climate Dynamics.

**We thank the reviewer for their positive and constructive comments.**

**Major comments:**

1) Introduction: Overall, the introduction is extensive and contains the relevant literature. I appreciate this very much! However, the introduction could become more streamlined through restructuring parts of it. In some cases, information is given that only fits the context to a limited extent (l. 60-61), or information is given too late in the text. For example, SAM is introduced in l. 72 but the information that SAM's negative phase is associated with an equatorward shift of the eddy-driven jet is only provided in line 94. To guide the reader a bit more, one could also list the main drivers (ENSO, IOD, SAM) in an introductory sentence and explain that the link to Australian spring temperatures is provided in the following. Further, the drivers are introduced in the order ENSO, IOD, SAM. However, when discussing the connection of heat conditions during spring (line 107 and after), the order is SAM, IOD, ENSO. As suggested before, please double-check the structure of the introduction and make sure that it follows a logic structure. I am convinced that the readership will appreciate this.

Thank you for this feedback. The introduction has been streamlined as suggested and also shortened in response to the other reviewers comments. Hopefully it is easier to follow and contains only the information required to follow the remainder of the paper.

2) Numerous studies emphasize the importance of land-atmosphere interaction for the occurrence of heat waves/heat extremes (e.g., Fischer et al. 2007; Hirsch et al. 2019) and the authors actually discuss this aspect in Section 7. However, the regression models to reconstruct monthly maximum temperature anomalies introduced in this study are purely based on oceanic/atmospheric predictors. By what degree would these models improve if additional predictors representing the land surface conditions (e.g., soil moisture) were considered. I strongly encourage the authors to conduct an analysis in this direction since it would (1) help to quantify the percent variance explained by the land-surface conditions, (2) raise interesting questions concerning the importance of the antecedent weather conditions in winter and the resulting moisture, and (3) potentially identify important sources of subseasonal predictability which is one of the motivating factors for this study.

Thank you for this useful comment suggesting we highlight the importance of antecedent soil-moisture to the development of anomalous heat through spring. As noted by the reviewer, other studies have looked at this extensively and the goal of this paper was to focus on the components of heat generation that are most directly associated with the atmospheric circulation (dynamical rather than thermodynamical). However, as suggested, we have done some preliminary investigation of the importance of antecedent moisture to spring monthly heat (added as Sup. Fig. 8, below). The statistical model of Australian weighted area-averaged maximum temperature was recalculated using the preceding month's rainfall anomaly to represent the antecedent moisture (e.g. Hope & Watterson, 2018), in addition to the three dynamical heat mechanisms. While the statistical model in September was not greatly improved, there is improvement later in spring (percent variance explained using the three dynamical mechanisms only was $r^2 = 0.81, 0.41, 0.59$ changed to $r^2 = 0.83, 0.68, 0.73$ with the inclusion of antecedent rainfall for September, October, and November respectively) As such, October rainfall may be a good source of predictability of November heat, and may be an interesting area for future research.

This preliminary result has been added to the discussion in the manuscript, along with a brief note in the introduction highlighting that we focus on the dynamical aspects of heat.

[Figure]

Figure S8. Australian weighted area-averaged maximum temperature statistically reconstructed using the three dynamical heat mechanisms (anomalous zonal wind, anomalous meridional wind (x-1), 500hPa vertical motion) and the previous month's Australian weighted area-averaged precipitation anomaly for September, October and November (blue bars) between 1979 to 2019. The red dashed bars show the observed temperature anomaly for each month. The dotted line shows the 95% confidence interval. The percent variance explained ($r^2$) for each month is in the top right of each figure.

**Minor comments:**

Title: From my point of view the title is not clear. It could mean that the tropical influence strengthens in a changing climate or during spring. According to my understanding, the latter is meant. To make this more clear the authors could write "Tropical influence on heat generating circulation over Australia strengthens in the course of spring".

**Agreed. The title has been changed to "Tropical influence on heat-generating atmospheric circulation over Australia strengthens through spring".**

l. 15: Please consider to specify "develop in a warming climate". I guess this is the meaning here.

**Agreed and changed.**

l. 60-61: This sentence appears to me a bit out of context. The sentence before and after describe the relation of ENSO and IOD to Australia's drier conditions and temperature. So, why is it important here to mention that ENSO and IOD co-vary?

**The intention was to lead into why using the TPI is useful to cover both tropical Pacific and Indian Ocean variability. The text has been modified to more clearly reflect this.**

l. 91: I guess it is not ENSO, IOD and the TPI in general that promotes anomalously high geopotential height. Please specify the phase that is referred to here (El Nino vs La Nina, positive/negative IOD etc).

**Yes. Corrected**

l. 100: Do the authors mean "dryness"?

**Yes. Corrected.**

l. 151: Please note that ERA5 is available from 1950 to almost present. I guess the authors are saying that in this study ERA5 data are used for the period 1979-2019. Please clarify.

**Yes. Only data from 1979-2019 were used. Corrected.**

l. 153: In Latex, please insert \, between number and unit, e.g., 500\,hPa.

**Corrected. We have added a space between the number and unit.**

l. 193: Please write "a" (radius of the earth) in math mode ($a$).

**Corrected.**

l. 199: Please double-check this sentence. According to Takaya and Nakamura, WAF is parallel to the local three-dimensional group velocity of a Rossby wave packet. In my view, the WAF itself is not propagating.

**Corrected.**

l. 229 and elsewhere: Also here, please check the meaning of the sentence. To my understanding one should rather say "... with the wave activity flux predominantly indicating a Rossby wave propagation from the subtropical Indian Ocean...".

**Agreed, and apologies for the inaccurate language. The text has been corrected throughout to better reflect how wave activity flux relates to Rossby wave propagation.**

l. 296: Are these winds also southerly when not considering anomalies but the actual wind? Caution is required when interpreting wind anomalies. If the actual wind is also southerly this could be mentioned in the text in support of the interpretation.

**Thank you for highlighting this point. The wind in the text and in the figures is the wind anomaly and is associated with the atmospheric circulation anomalies. The text has been corrected throughout to clarify that we are referring to anomalies, not the actual wind.**

**To further aid interpretation, the climatological monthly wind for Australia is available at http://www.bom.gov.au/jsp/ncc/climate_averages/wind-velocity/index.jsp?period=sep#maps In general, wind in early spring is from the west across extratropical Australia and from the east in the lower latitudes. By November the subtropical ridge has shifted poleward such that mean east to southeasterly winds extend across much of the continent. As such, a southerly wind anomaly associated with the TPI would be associated with greater cold-air advection and reduced maximum temperatures in early spring than average. A brief comment about the spring climatological winds has been added to the introduction.**

l. 381: Are you saying that WAF indicates a Rossby wave propagation along the jet waveguide from Africa or is it to Africa? This would be difficult to see since Africa is hardly shown on the map. Please clarify.

**Apologies for this typo. It should have been 'from' not 'to'. Text amended.**

l. 388: Better provide the information on what "waveguide" is referring to when it is first introduced in the text.

**Agreed. Text amended to introduce waveguide and add reference**

l. 516: That much of the atmospheric patterns associated with heat through spring are explained by neither the tropical TPI nor SAM is also related to my major comment #2. Would land-atmosphere processes explain at least parts of it?

**That is an interesting idea, and land-atmosphere processes may form part of the story (see response to major comment #2). Part of the original motivation for this study was expecting that the major climate drivers (ENSO, SAM, TPI, IOD etc) would only explain a relatively small percentage of heat as they are simple indices representing complex systems remote from Australia. Australian temperature anomalies are a combination of many different factors, only some of which will be related to the TPI or SAM, as our study has now quantified.**

l. 554-557: This is an interesting thought. Quinting and Reeder (2017) actually show that air masses ending up in upper-tropospheric anticyclones during heat waves are strongly diabatically heated during their ascent south of Australia. So, their results potentially support the hypothesis given here concerning the storminess just to Australia's south.

**Agreed and thank you for the comment. Their results are what made us think this is what is happening in this month-long spring example. It would be very interesting to explore this idea further in future work to understand whether this mechanism is applicable on a month- to season- time scale, rather than the synoptic scale used in Quinting and Reeder (2017) and how applicable it is outside of the austral summer season that they examined. The text has been adjusted slightly to better reflect Quinting and Reeder's work as the inspiration for this idea.**

Figure 1: To better match the order of the remote drivers in the text, my suggestion is to reorder the panels of Fig. 2: SAM, Nino, DMI, TPI. In the caption of the figure, please correct units (e.g., hPa\,s$^{-1}$). The caption was probably rendered incompletely as its last sentence ends without a subject.

**We have added spaces in the units as suggested. We apologise that the caption was incomplete, it has now been corrected.**

**While we appreciate the suggestion, we have not reordered the figure as suggested as the remainder of the paper focuses on SAM and the TPI. Instead, the text (lines 467-469) has been readjusted to discuss TPI before ENSO and IOD.**

Figure 4: I guess it is Ks instead of K. Also, how exactly are the monthly mean climatological Ks calculated. Are these again obtained through linear regression? Please explain in the text.

**Apologies for the incomplete caption. K should have been Ks. The figure is the climatological stationary wave number calculated using 1981-2010 climatological wind for September, October, and November. The caption has been updated with this information.**

**Technical Notes:**

l. 73: Please place comma after the references.

**The references have been corrected.**

l. 75: I guess "to" is not needed here.

**Agreed. Corrected.**

l. 97: I guess it should be "deflecting" instead of "defecting".

**Corrected.**

l. 165: "impact" instead of "impacts".

**Corrected**

l. 254: I guess "of" is missing between "poleward" and "that".

**Corrected**

l. 305: I assume that "appears" needs to be deleted.

**Corrected**

l. 320: Insert "to" between "relative" and "that".

**Corrected**

l. 569: "associated" instead of "associate".

**Corrected**

Figure 2: "Area-averaged" instead of "areal-averaged".

**Corrected**

**References:**

Fischer, E. M., Seneviratne, S. I., Lüthi, D., and Schär, C. (2007), Contribution of landatmosphere coupling to recent European summer heat waves, Geophys. Res. Lett., 34, L06707, doi:10.1029/2006GL029068.

Hirsch, A. L., Evans, J. P., Di Virgillio, G., Perkins-Kirkpatrick, S. E., Argüeso, D., Pitman, A. J., et al (2019). Amplification of Australian heatwaves via local land-atmosphere coupling. Journal of Geophysical Research: Atmospheres, 2019; 124: 13625– 13647. https://doi.org/10.1029/2019JD030665

Quinting, J. F., & Reeder, M. J. (2017). Southeastern Australian Heat Waves from a Trajectory Viewpoint, Monthly Weather Review, 145(10), 4109-4125

**Reviewer #2**

The study provides an analysis of the austral spring Australian maximum temperatures and their relationship with the dynamics in the tropics and extratropics. Using statistical analysis on observations and reanalysis, the study finds that Australian maximum temperatures in early spring can be partially explained by variability in the mid-latitudes associated with the Southern Annular Mode, while at the end of the season they are more related to tropical variability. The findings of this study provide new insights into the connection between maximum temperatures in Australia and modes of variability. I recommend acceptance for publication after minor revisions.

**We thank the reviewer for their constructive comments.**

The introduction is quite long and broad; it can be shortened to keep to the point of the study. Cite only the studies that are essential for building the motivation and hypothesis of this study.

**The introduction has been streamlined as recommended.**

In methods, has the background warming trend been removed from temperature and other data detrended prior to the analysis?

**Yes. All the data, including the Australian maximum temperature time series, was linearly detrended prior to analysis. This was stated in the text at the end of the methods but has been moved to a separate line to improve clarity.**

Minor points:

L.139: Add a sentence to explain the aims of the study before listing what is presented in the next sections.

**Agreed. A line has been added as suggested.**

L.153: replace "omega" with 'velocity'

**Agreed. Word replaced as suggested.**

L.145,163,208: missing period at the end of the sentence.

**Corrected as necessary.**

L.1193: "'E'arth"

**Corrected.**

L.232: Typo in "patterns"

**Corrected.**

L.298: "patterns"

**Corrected.**

L.305: delete "appears"

**Agreed.**

L.516: The end of the sentence is missing.

**Apologies, a comma was in the place of a full stop. Corrected.**

L.557: Delete 'heat'

**Corrected**

Figure 1 caption: It seems that the end of the sentence was cut. Do you mean '39 independent samples'?

**Yes. Apologies the end of the caption was cut-off. Corrected.**

Figure 5 caption: Observed values are in 'pink' instead of blue?

**Yes. Corrected**

---

## Author Response (AR2)

Manuscript: WCD-2021-63

Strengthening tropical influence on heat generating circulation over Australia

through spring

by Roseanna C. McKay, Julie M. Arblaster, and Pandora Hope

**Author response to technical corrections**
The authors thank the two reviewers and the editor for their thorough evaluation of our manuscript. The suggested technical corrections have now been made and improved the manuscript as a result. The reviewers' and editor's comments are in bold and our responses are in plain text.

**Editor**

**line 188: missing period**
Corrected

**line 577, and throughout manuscript: instead of "heat mechanisms", I would suggest "heating mechanisms"**
Changed throughout

**check throughout manuscript, e.g. lines 218, 582, 504: "SAM" -> "the SAM"**
Changed throughout

**line 216: clarify "(x-1)" when using it the first time**
Added "(denoted by x-1)" to line 222 (216 in previous version)

**lines 495/496: "correlate" with what? please specify**
Apologies, "with temperature" has been added

**line 500: missing verb**
Corrected to include, 'is'. Sentence structure has also been changed for clarity.

**line 502: "the tropics promotes" -> "the tropics promote"**
Corrected

**line 510 - 511: please check grammar**
Sentence has been changed to, **"**However, the upper-level atmospheric anomalies through spring are broadly consistent with IOD-forced wave trains identified in the literature, with shifts poleward or equatorward as discussed above (Cai et al., 2011; McIntosh and Hendon, 2018; Wang et al., 2019)." On lines 561-564 of edited document.

**line 519 - 520: please check grammar**
Sentence has been changed to "To test this idea, wave activity flux is calculated from regressions of the three dynamical heating mechanisms onto 200Z (Sup. Fig. 7). As spring progresses, WAF associated with the heating mechanisms indicates greater Rossby wave

propagation out of the tropics than the extratropics. Waves appear to propagate along the jet waveguide toward Australia, reflecting the growing relationship between the tropics and temperature." On lines 568 to 574 of edited document.

**line 557: missing space**
Corrected

**Review 1**
**The authors addressed all my previous comments satisfactorily.**
**The introduction has been shortened and readability improved.**
**Apologies to have missed the information that data was detrended. There is no need to move that sentence to a different paragraph. It can be merged with the previous sentence (L.159).**
Sentence has been combined with following sentence to create a single short paragraph instead of a single sentence.
**Please revise the text to add final '.' at the end of sentences, e.g. L.141, L.159, L.188, Fig.2 caption.**
Corrected.
**The manuscript can then be accepted for publication in my view.**

**Review 2**
**Minor/Technical:**
**l. 141: Missing full stop.**
Corrected

**l. 159: Missing full stop. Please consider to join the single-sentence paragraphs to one paragraph.**
Corrected and two short paragraphs combined into one.

**l. 355: K_s instead of Ks.**
Changed

**l. 374 and elsewhere: K_s instead of Ks.**
Changed.

**l. 477: Please revise this sentence: "...are influenced differently by the different influence..." reads somewhat awkward.**
Agreed and changed to "Overall, it appears that the heating mechanisms associated with high maximum temperatures in spring are subject to the changing influence of the extratropics and tropics on the local atmospheric circulation features through spring." On lines 512 – 515 of edited document.

**l. 489: Please specify which oceans you are referring to. I guess it is the subtropical ocean regions? At least advection of air from the Southern ocean would still lead to a cooling.**
Agreed and changed to 'subtropical' oceans

**Figure captions and text: Please double-check format of SI units. Quite often a space is missing. E.g., ms^{-1} should be replaced by m\,s^{-1}.**

Corrected